# The Epigenetic Regulation of RNA N6-Methyladenosine Methylation in Glycolipid Metabolism

**DOI:** 10.3390/biom13020273

**Published:** 2023-02-01

**Authors:** Haiqing Yang, Yuting Li, Linying Huang, Miaochun Fang, Shun Xu

**Affiliations:** Guangdong Provincial Key Laboratory of Medical Molecular Diagnostics, Institute of Aging Research, School of Medical Technology, Guangdong Medical University, Songshan Lake, Dongguan 523808, China

**Keywords:** N6-methyladenosine methylation, glucose metabolism, lipid metabolism, glycolipid metabolic disease

## Abstract

The highly conserved and dynamically reversible N6-methyladenine (m6A) modification has emerged as a critical gene expression regulator by affecting RNA splicing, translation efficiency, and stability at the post-transcriptional level, which has been established to be involved in various physiological and pathological processes, including glycolipid metabolism and the development of glycolipid metabolic disease (GLMD). Hence, accumulating studies have focused on the effects and regulatory mechanisms of m6A modification on glucose metabolism, lipid metabolism, and GLMD. This review summarizes the underlying mechanism of how m6A modification regulates glucose and lipid metabolism-related enzymes, transcription factors, and signaling pathways and the advances of m6A regulatory mechanisms in GLMD in order to deepen the understanding of the association of m6A modification with glycolipid metabolism and GLMD.

## 1. Introduction

M6A has been established to be a reversible RNA methylation modification, which exerts critical roles in the post-transcriptional regulation of gene expression [1,2]. M6A methylation is the most prevalent and internal chemical modification in eukaryotic messenger RNA(mRNA) and long non-coding RNAs (lncRNAs) and is highly conserved among species [3], which is enriched in the stop codon, the 3′-untranslated region (3′UTRs), or the long internal exon, and usually occurs in the consensus motif of RRACH ([G/A/U] [G > A] m6 AC[U > A > C]) [4]. Functionally, m6A methylation is widely implicated in RNA metabolism by affecting RNA maturation, splicing, folding, export, localization, translation efficiency, and stability [5,6,7,8,9,10] and thus is involved in various biological processes.

Metabolic diseases have increasingly become a severe problem for the global healthcare system and arise from various risk factors, including abnormal glycolipid metabolism. Extensive studies have demonstrated that abnormal glycolipid metabolism is closely associated with glucolipid metabolic diseases (GLMD), such as obesity, diabetes mellitus, hyperlipidemia, non-alcoholic fatty liver disease, hypertension, and atherosclerosis [11]. Accumulating evidence unveiled that m6A methylation exerted crucial effects on nutritional physiology and metabolism, and its dysregulation caused alterations in the circadian rhythm, metabolic pathway, inflammatory state, and cancer progression [12]. Therefore, a probe into the effect and underlying mechanism of m6A methylation on glycolipid metabolism and GLMD will not only deepen the understanding of the relationship between glycolipid metabolism and GLMD but also provide a novel strategy for the diagnosis and therapy of GLMD.

Recent studies have identified numerous m6A-regulated genes, including glucose and lipid metabolism-related enzymes, transcription factors, and signaling pathways, which exerted important effects on hyperglycemia, insulin signaling transduction, lipid accumulation, and plaque formation. We herein reviewed the m6A regulatory function in glycolipid metabolism and glucolipid metabolic disorders in order to provide insights into the regulation of glycolipid metabolism homeostasis and the clinical diagnosis and treatment of GLMD.

## 2. m6A Methylation

M6A mainly refers to the methylation of the sixth nitrogen of adenosine, which accounts for the most abundant internal modification that occurs in eukaryotic mRNA. Other than mRNA, m6A occurs in long non-coding RNA (lncRNA), circular RNA (circRNA), and microRNA (miRNA) [13,14,15]. The highly conserved and dynamic m6A modification level has been established to be regulated by methyltransferases (termed “writers”) and demethylases (termed “erasers”) (Table 1). The m6A writers comprised Methyltransferase-like protein 3 (METTL3), methyltransferase-like protein 14 (METTL14) [16], methyltransferase-like protein 16 (METTL16), Wilms tumor 1-associated protein (WTAP), Vir-like m6A methyltransferase associated (VIRMA/KIAA1429), RNA-binding motifs protein15/15B (RBM15/15B), and Zinc Finger CCCH-Type Containing 13 (ZC3H13). Typically, METTL3, METTL14, and WTAP form a methyltransferase complex (MTC), which recognizes a consensus RNA sequence RRACH (R = G or A; H = A, C or U) and catalyzes the m6A modification on mRNAs, while METTL16 regulates S-adenosylmethionine (SAM) homeostasis [17]. VIRMA recruits MTC to specific mRNA regions and interacts with cleavage and polyadenylation specific factor 5/6 (CPSF5/6) [18]. RBM15/15B mainly guides METTL3-METTL14 heterodimer to uracil U-rich RNA sites for methylation [19]. ZC3H13 bridges WTAP to the mRNA binding factor Nito and contributes to the nuclear localization of MTC [20]. Additionally, IME4 and MUM2 mediate m6A modification of yeast mRNA [21].

The discovery of m6A demethylases, including fat mass and obesity-associated (FTO) and ALKBH5, verify m6A methylation as a dynamic and reversible RNA modification and thus are regarded as m6A erasers. Both FTO and ALKBH5 belong to the alpha-ketoglutarate-dependent dioxygenase family. FTO promotes mRNA splicing and translation [22], while ALKBH5 mainly promotes mRNA nuclear export, mRNA splicing, and long 3′-UTR mRNA production by clearing m6A [23].

The m6A binding proteins, referred to as “Readers”, specifically recognize and bind with the m6A-modified mRNA to regulate gene expression via impacting mRNA transcription, stability, splicing, or nuclear export. The most important m6A readers are YTH domain-containing family proteins, including YTHDF1/2/3 and YTHDC1/2. YTHDF1 promotes mRNA translation and protein synthesis, and YTHDF2 reduces mRNA stability and regulates mRNA localization; YTHDF3 interacts with YTHDF1 to promote mRNA translation or assists YTHDF2-mediated RNA degradation [24]. Nuclear YTHDC1 mediates RNA splicing, export, and transcriptional silencing [25]. YTHDC2 mainly promotes the translation of target RNA but reduces their abundance [26]. In addition, eukaryotic translation initiation factor 3 (eIF3) promotes mRNA translation by recruiting ribosomal initiation complexes [27]. Insulin-like growth factor 2 mRNA binding proteins (IGF2BPs) enhance the stability of target transcripts, and the heterogeneous nuclear ribonucleoprotein (HNRNP) family mainly mediates mRNA splicing [28,29]. Among these, HNRNPA2B1 regulates alternative splicing and primary microRNA processing, and HNRNPC/G mediates pre-mRNA splicing and processing.

## 3. m6A Modification and Glucose Metabolism

Glucose metabolism involves a very complex regulatory network, including anaerobic glycolysis, aerobic oxidation, pentose phosphate pathway, glycogen synthesis, and gluconeogenesis [30]. An increasing number of studies have reported that m6A modification is an important regulatory mechanism of glucose homeostasis and downstream effects (Figure 1).

### 3.1. Glycolysis

Glycolysis is a pivotal energy-producing pathway in organisms, which decomposes glucose to pyruvate under anaerobic conditions, with the release of free energy into adenosine triphosphate (ATP). Typically, monosaccharides are transported to the cytoplasm by glucose transporters (GLUTs) or sodium-dependent glucose cotransporters (SGLTs), then undergo the preparation phase of glucose activation and cleavage and the release energy phase of oxidative phosphorylation, which are closely related to hexokinase (HK), phosphofructokinase-1 (PFK-1) and pyruvate kinase (PK), and ultimately generate pyruvate and ATP [31].

A growing number of studies have uncovered the broad effects of m6A modification on metabolic networks by regulating glycolytic genes and signaling pathways. METTL3/IGF2BP2-mediated m6A modification has been reported to promote glycolysis by enhancing the stability of HK2 and GLUT1 [32,33]. METTL14-mediated m6A modification not only promotes glycolysis by attenuating sirtuin6 (SIRT6) stability but also facilitates hypoxia-inducible factor 1 subunit alpha (HIF1A)-mediated glycolysis and cell proliferation by inhibiting the expression of phosphatase LHPP [34,35]. WTAP enhances glycolytic activity by mediating m6A methylation of HK2 and enolase 1 (ENO1) mRNA [36,37,38]. The m6A methyltransferase KIAA1429 positively regulates aerobic glycolysis in a GLUTI or HK2-dependent manner [39,40], and ZC3H13 facilitates glycolysis by enhancing the stability of pyruvate kinase M2 (PKM2) mRNA [41]. In addition, FTO-mediated demethylation promotes HK2 and PKM2-mediated glycolysis via upregulating the expression of lncRNA HOTAIR and autophagy associated 5 gene (ATG5), respectively [42,43], while down-regulation of FTO participates MYC-mediated cellular glycolysis and the regulation of IL-6/JAK2/STAT3 signaling pathways [44,45]. FTO/YTHDF2 mediates post-transcriptional upregulation of phosphofructokinase platelet (PFKP) and lactate dehydrogenase B (LDHB) and activates aerobic glycolysis [46]. ALKBH5 is involved in the regulation of casein kinase 2 (CK2) α-mediated glycolysis in an m6A-dependent manner [47]. M6A reader IMP2 enhances the stability of lncRNA ZFAS1 and facilitates the exposure of ATP-binding sites, thereby accelerating ATP hydrolysis and glycolysis [48]. IGF2BP1/2 is involved in the regulation of MYC-mediated glycolysis and provides an additional energy source for cell metabolism [49,50,51]. In cellular metabolism, pyruvate dehydrogenase kinase 4 (PDK4) methylation can be recognized by YTHDF1/eEF-2 complex and IGF2BP3 to direct carbon flux from oxidative phosphorylation (OXPHOS) to glycolysis [52]. The interaction of YTHDF2 with RNA-binding motif protein 4 (RBM4) promotes signal transduction and activator of transcription 1 (STAT1)-mediated glycolysis, which participates in regulating macrophage polarization and inflammatory factor expression [53].

### 3.2. Pentose Phosphate Pathway

The pentose phosphate pathway (PPP), also known as the hexose phosphate bypass, is generally divided into two branches: oxidative and non-oxidative. During the highly active oxidation phase in most eukaryotes, glucose-6-phosphate (G-6-P) is converted to ribulose-5-phosphate, carbon dioxide, and nicotinamide adenine dinucleotide phosphate (NADPH) [54]. The non-oxidative branch is nearly ubiquitous and supports the nucleic acid skeleton and aromatic amino acid biosynthesis by increasing the expression of 5-phosphate ribose and erythritol-4-phosphate [55,56].

As a key enzyme of PPP, glucose-6-phosphate dehydrogenase (G6PD) is overactive in metabolic pathways and participates in the regulation of redox status. Recent studies have demonstrated that tumor cells respond to chemotherapy-induced reactive oxygen species (ROS) accumulation by activating PPP to increase NADPH, thereby adapting to oxidative stress and maintaining malignant cell proliferation [57]. In a glioma, ALKBH5 enhances the stability of G6PD mRNA and PPP flux by eliminating m6A methylation [58]. YTHDF2 facilitates mRNA translation of G6PD in an m6A-dependent manner, thereby enhancing PPP activity and tumor cell viability [59]. In addition, YTHDF2 promotes the miR-663b/DLG4/G6PD axis and pentose phosphate pathway by mediating circ_0003215 RNA degradation [60].

### 3.3. Glycogen Synthesis and Gluconeogenesis

As a storage form of sugar, glycogen synthesis refers to the process of converting activated glucose into glycogen under the catalysis of glycogen synthase. Glycogen synthase activity is regulated by phosphorylation/de-phosphorylation of various serine/threonine kinases, among which glycogen synthase kinase 3 (GSK-3) is widely involved in the physiological and metabolic processes. Under normal feeding conditions, enhanced insulin signaling activates protein kinase B (AKT), which then inactivates GSK-3 through phosphorylation and ultimately promotes glycogen synthesis in response to increased glucose uptake. In contrast, fasting activates GSK-3 by de-phosphorylation, which inhibits glycogen synthesis and facilitates glycogenolysis, supplying the body with fuel reserve [61]. As a highly conserved negative regulator of receptor tyrosine kinases, cytokines, and Wnt signaling pathways, GSK-3 participates in the m6A methylation regulatory network and affects multiple downstream effectors. Similarly, downregulated microRNA-6125 and hypoxia-induced lncRNA STEAP3-AS1 interact competitively with YTHDF2, which results in phosphorylation and inactivation of GSK3β, activation of Wnt/β-catenin signaling pathways, and glycogen synthesis [62,63]. METTL14 negatively regulates the expression of fibroblast growth factor receptor 4 (FGFR4) in an m6A-dependent manner, while FGFR4 activates glycogen synthesis and the β-catenin/TCF-4 signaling pathway through phosphorylation of GSK-3β [64]. M6A mediates phosphorylation of AKT/GSK-3β and activation of tensin homolog protein (PTEN), which promotes glycogen synthesis and protects neurons from pyroptosis induced by cerebral ischemia/reperfusion (I/R) [65]. Cardiac hypertrophy-associated PIWI-interacting RNA (CHAPIR) competitively binds METTL3 and blocks m6A methylation of polymerase family member 10 (PARP10), while up-regulation of PARP10 facilitates glycogen synthesis and pathological cardiac hypertrophy by inhibiting the kinase activity of GSK-3β [66].

The process by which organisms synthesize glucose or glycogen from non-sugar precursors such as lactic acid, glycerol, and glycogenic amino acids is known as gluconeogenesis. Under starvation, the liver promotes gluconeogenesis by decreasing insulin concentration and increasing glucagon concentration, which is the main cause of diabetic hyperglycemic phenotype [67]. METTL14 deficiency leads to decreased β-cell mass and insulin secretion, but β-cell-specific knockdown of METTL14 enhances insulin signaling and reduces hepatic gluconeogenesis under a high-fat diet (HFD), thus improving insulin sensitivity with compensated [68]. This suggests that m6A methylation regulates the gluconeogenic flux in response to insulin signaling, which is of great significance for glucose homeostasis.

## 4. m6A Modification and Lipid Metabolism

Lipids are essential structural components of the cell membrane, including fats, phospholipids, sphingolipids, and cholesterol lipids, which not only serve as molecular signals and energy sources but also participate in metabolism [69]. Recently, a growing body of studies has focused on the association of m6A methylation with triglyceride metabolism, cholesterol metabolism, and plasma lipoprotein metabolism, and thus, we herein summarize the regulatory mechanism of m6A modification in lipid accumulation and downstream effects (Figure 2).

### 4.1. Triglyceride Metabolism

As a significant form of energy storage and oxidative energy supply, triglyceride (TG) is mainly synthesized from glycerol and fatty acids (FA) provided by glucose metabolism. FA is cleared mainly through intracellular β-oxidation or TG-rich very low-density lipoprotein (VLDL) entering the blood; thus, TG represents the main storage and transport form of FA in the cell and plasma [70]. FTO-mediated demethylation reduces the mitochondrial content of hepatocytes and promotes TG accumulation, suggesting that m6A modification is linked to fat metabolism [71]. FTO/YTHDC2 activates the transcription of thermogenesis genes and facilitates the browning of white fat by promoting the expression of HIF1A, which is conducive to fighting obesity by increasing energy expenditure [72]. FTO/YTHDF2 inhibits autophagy and lipogenesis by mediating mRNA degradation of autophagy-related genes ATG5 and ATG7 [73]. Metabolism is usually regulated by RNA transcription and translation, but some metabolites, such as NADP, can regulate m6A demethylation and lipogenesis by directly binding to FTO [74].

In addition, METTL3-mediated m6A modification promotes the expression of fatty acid synthase (FASN) and fatty acid metabolism, which is involved in the regulation of blood glucose homeostasis and insulin sensitivity [75]. FTO/YTHDF2 mediates mRNA degradation of FASN, while the low expression of FASN reduces lipid accumulation through inhibition of de novo fat synthesis [76]. Overexpression of METTL3 leads to shortened RNA half-lives of metabolism-related genes, which aggravates HFD-induced lipid metabolism disorder and insulin resistance [77]. YTHDC2 inhibits TG accumulation by reducing the mRNA stability of lipogenesis genes, thereby improving hepatic steatosis and insulin resistance [78].

Lipids drive energy metabolism, inflammatory signaling, and immune mechanisms, while FA participates in the activation and regression of inflammatory reactions. In fatty acid-induced cardiomyocyte inflammation, FTO-mediated demethylation increases the serum levels of total cholesterol (TC), TG, and low-density lipoprotein cholesterol (LDL-C) by improving the expression of differentiation 36 (CD36), which leads to lipid deposition and myocardial injury [79]. In an intestinal inflammatory response, METTL3 mediates m6A methylation of TNF receptor-associated factor 6 (TRAF6) and promotes its transfer from the nucleus to the cytoplasm, thereby activating NF-κB and mitogen-activated protein kinase (MAPK) signaling pathways, which finally results in malabsorption of long-chain fatty acids (LCFAs) and TG accumulation [80]. M6A epigenetic modification also regulates lipid metabolism and pharmacokinetics by affecting the mRNA stability of carboxylesterase 2 (CES2) [81]. M6A-induced lncDBET activates the PPAR signaling pathway and lipid metabolism through direct interaction with FABP5, while lncNEAT1 and HNRNPA2B1-mediated RPRD1B stability facilitate fatty acid uptake and synthesis via c-Jun/c-Fos/SREBP1 axis [82,83].

### 4.2. Cholesterol Metabolism

Cholesterol maintains the integrity and fluidity of biofilms, which depend on the regulation of its synthesis, uptake, efflux, esterification, transformation, and transport [84]. In cholesterol homeostasis, sterol regulatory element-binding proteins (SREBPs) promote endogenous synthesis and exogenous uptake of cholesterol, while hepatic X receptors (LXRs) induce cholesterol efflux [85,86].

METTL14 limits cholesterol efflux and promotes atherosclerotic plaque inflammation by mediating the m6A modification of lncRNA ZFAS1 [87]. On the one hand, FTO-mediated demethylation inhibits lipolysis and promotes the development of obesity through the sterol regulatory element-binding protein 1 c (SREBP1c) pathway, but on the other hand, it accelerates cholesterol efflux and inhibits atherosclerosis development through phosphorylation of AMPK [88]. METTL3/14 in a non-alcoholic fatty liver disease (NAFLD) model increased the m6A modification and protein level of ATP citrate lyase (ACLY) and stearoyl-CoA desaturase 1 (SCD1), which promoted cholesterol production and lipid droplet deposition [89]. Overexpression of YTHDF2 accelerates m6A-mediated mRNA degradation of LXRA and HIVEP zinc finger 2(HIVEP2), which affects cholesterol synthesis, efflux, and uptake [90].

### 4.3. Plasma Lipoprotein Metabolism

Plasma lipoproteins are divided into chylomicrons (CM), VLDL, low-density lipoproteins (LDL), and high-density lipoproteins (HDL) according to their density. Among these, CM mainly transports exogenous TG and cholesterol, VLDL transports endogenous TG, LDL transports endogenous cholesterol, and HDL is characterized by reverse cholesterol transport [91].

The specific binding of an anthraquinone compound LuHui Derivative (LHD) to FTO inhibits the expression of CD36, which attenuates the inflammatory response and improves cardiac function through the reduction of plasma lipoprotein [79]. Exploring the epigenetic regulation of m6A modification and lipoprotein metabolism in the model of oxidized low-density lipoprotein (ox-LDL)-induced atherosclerosis (AS) has become an important research area in recent years. METTL3 mediates vascular smooth muscle cell (VSMC) phenotypic transformation and stabilizes AS plaques by promoting plasma lipoprotein metabolism in an ox-LDL-induced VSMC AS model, while METTL14 facilitates lipoprotein synthesis and AS development by mediating the m6A modification of p65 mRNA in human umbilical vein endothelial cells (HUVEC) [92,93]. Another mechanism suggested that METTL3 promotes ox-LDL-mediated lipoprotein metabolism disorders and macrophage inflammation through the activation of STAT1 signaling [94]. These results demonstrated that m6A modification is involved in the regulation of inflammation mediated by plasma lipoprotein metabolism.

## 5. m6A Modification in Glucolipid Metabolic Disease (GLMD)

GLMD, including obesity, diabetes, hyperlipidemia, non-alcoholic fatty liver disease, hypertension, and atherosclerosis, is characterized by single or combined disorders of glucose and lipid metabolism, which arise from multiple risk factors, such as insulin resistance, oxidative stress, chronic inflammatory reaction, neuroendocrine dysfunction, and intestinal microbiota imbalance [95]. As an important regulatory mechanism in glycolipid metabolism, m6A modification is closely related to the complexity and systemic nature of GLMD (Figure 3).

### 5.1. Obesity

Obesity is mainly manifested as high overall or local body fat content and ectopic fat deposition, which arises from a long-term imbalance between energy intake and body consumption. The gene expression profile of human adipose tissue indicated that several m6A modification-related genes, including WTAP, VIRMA, ALKBH5, and YTHDC1, are correlated with obesity, body mass index (BMI), and clinical variables, while the single nucleotide polymorphisms (SNPs) of METTL3 and YTHDF3 are associated with anthropometric and metabolic variables [96]. This suggested a potential role for m6A modification in obesity in spite of the limitation by individual differences and tissue specificity.

Numerous studies have stated that FTO is associated with obesity in the form of m6A demethylase and plays an important role in lipogenesis and obesity susceptibility. Individuals with FTO risk variants have higher body weight, fat mass, and BMI due to increased hunger and energy intake [97,98]. FTO-mediated demethylation inhibits the recruitment of runt-related transcription factor 1 (RUNX1T1) by the splicing regulatory (SR) protein SRSF2, then induces the differentiation of mouse 3T3-L1 preadipocytes by regulating the alternative splicing of the adipose gene RUNX1T1 [99]. Furthermore, FTO increases the expression of peroxisome proliferator-activated receptor gamma (PPARG) mRNA by downregulating its m6A level, which promotes the differentiation of bone marrow stem cells (BMSCs) into adipocytes [100]. In turn, obesity induces glycolysis, lipid toxicity, and pro-inflammatory phenotype by increasing FTO protein expression [101]. YTHDF1 facilitates mitochondrial carrier 2 mRNA translation and participates in lipogenesis in an m6A-dependent manner [102], and YTHDF2 plays a role in mitotic clone expansion and obesity by mediating mRNA decay of adipogenic regulators [103]. LncRNA NEAT1 acts as an m6A-modified transcript, which is regulated by miR-140 in adipogenesis [104]. In summary, m6A modification is involved in the regulation of obesity-related biological processes such as adipogenesis and lipid metabolism.

### 5.2. Diabetes Mellitus (DM)

DM occurs mostly in the context of β-cell dysfunction and insulin resistance due to the feedback loop between insulin action and insulin secretion not working properly [105]. Accumulating evidence has unveiled that m6A modification exerts a critical role in the expression of key regulatory genes through multiple effects during diabetes.

The pathogenesis of diabetes is very complex, involving glucose homeostasis, lipid metabolism, and β-cell biology. The results of liquid chromatography/electrospray ionization/tandem mass spectrometry revealed that high FTO expression in type 2 diabetes mellitus (T2DM) patients accompanied by increased mRNA expression levels of key genes of glucose metabolism (FOXO1, G6PC, and DGAT2) [106]. METTL3 promotes fatty acid metabolism and insulin resistance by mediating m6A methylation of FASN [75], while YTHDC2 plays an inhibitory role in lipid accumulation [78]. Consistently, IMP2/IGF2BP2 coordinate IGF2-AKT-GSK3β-PDX1 signaling in an m6A-dependent manner, which influences insulin secretion and T2DM susceptibility [107]. Moreover, METTL3/14 modulates β-cell proliferation and functional maturation during early islet development in neonatal mice [108], whose depletion is associated with β-cell failure, impaired insulin secretion, and glucose intolerance in the progression of diabetes [109]. High glucose concentrations in human and mouse islets induce low levels of m6A, while diabetes-induced PARP1 expression is regulated by YTHDF2-mediated m6A modification [110]. METTL14/YTHDF2 suppresses pyroptosis and diabetic cardiomyopathy by mediating the degradation of LncRNA TINCR, while LncRNA Airn alleviates diabetic cardiac fibrosis via IMP2-p53 axis in an m6A-dependent manner [111,112]. METTL3 regulates endothelial-mesenchymal transition in diabetic retinopathy via lncRNA SNHG7/KHSRP/MKL1 axis [113].

### 5.3. Hyperlipemia

Hyperlipidemia mainly refers to the disorder of plasma lipoprotein, including hypercholesterolemia, hypertriglyceridemia, low high-density lipoproteinemia, and mixed hyperlipidemia [114], which is closely associated with cerebral infarction, coronary artery disease (CAD) and atherosclerosis.

In HFD mice and palmitate (PA)-induced C2C12 cells, METTL3 induced oxidative stress and impaired glucose uptake by inhibiting the mRNA stability of the serine-threonine kinase protein kinase D2 (PRKD2) [115], which conferred the models with GLUT4/IRS-1/AKT signaling inhibition and high levels of glucose and TG. Additionally, the m6A levels of cytokine signaling inhibitor 2 (SOCS2) are associated with increased cholesterol and TG in hepatic steatosis [116]. MicroRNAs (miR16-1-3p, miR101a-3p, miR362-3-5p, miR501-5p, miR532-3p, and miR542-3p) regulate m6A methylation in cholesterol efflux pathway [117]. Additionally, one group reported that resveratrol treatment increased the mRNA expression of PPARα, cytochrome P450, and fatty acid-binding protein 4 (FABP4) by reducing the m6A level in HFD mice, which led to significant reductions in serum TC, TG, and LDL contents [118]. Dyslipidemia mediates the suppression of the cAMP response element binding protein (CREB) pathway in the liver and the AMPK pathway in skeletal muscle, both of which are regulated by FTO-dependent demethylation [119,120]. These results demonstrate the important regulatory role of m6A methylation in plasma lipoprotein metabolism and hyperlipidemia.

### 5.4. Nonalcoholic Fatty Liver Disorders (NAFLD)

NAFLD is primarily divided into three stages: the first is pure fat accumulation in the liver of a nonalcoholic fatty liver (NAFL), then the liver cell damage (balloon sample change) and inflammation of nonalcoholic steatohepatitis (NASH), which finally leads to liver fibrosis, cirrhosis, and HCC [121]. The hepatocyte injury of NAFLD is mainly driven by the overload of metabolic substrates such as glucose and fatty acids [122]. Thus, the dysregulation of glucose and lipid metabolism are pivotal events in the development of NAFLD, which is characterized by necrotizing inflammation and liver fibrosis.

In a NAFLD mouse model and FFA-induced hepatocyte model, METTL3/YTHDF1 conjugation mediated m6A modification and promoted Rubicon mRNA expression, which led to hepatic lipid deposition [123]. In the glucocorticoid (GC)-induced NAFLD model, trans-activated FTO induced hepatocyte adipogenesis and lipid accumulation by mediating the demethylation of sterol regulatory element binding factor 1 (SREBF1) and SCD1 [124], while FTO promoted the progression of chronic liver inflammation by mediating demethylation of interleukin-17 (IL-17) [125], indicating the critical role of FTO in the development of NAFL and NASH to HCC. ALKBH5 promotes cullin4A (CUL4A)-linked degradation of inositol polyphosphate phosphatase-like 1 (INPPL1, SHIP2) by increasing the RNA stability of LINC01468, which drives lipogenesis and NAFLD-HCC progression [126]. What’s more, YTHDF3 directly regulates peroxiredoxin 3 (PRDX3) translation in an m6A-dependent manner, thereby reducing hepatic stellate cell (HSC) activation and liver fibrosis in response to mitochondrial oxidative stress [127]. Another study noted that ALKBH5 reduced type I collagen and α-smooth muscle actin (α-SMA) levels by activating PTCH1, which finally inhibits HSCs activation and liver fibrosis [128]. Hypermethylated transcripts of NAFLD are mainly enriched in lipid metabolism processes, and higher HDL cholesterol and unsaturated fatty acid proportions are accompanied by the dysregulation of MYC expression and m6A methylation [129,130]. To sum up, m6A modification mediates the pathogenesis of NAFLD by regulating hepatic lipid deposition, inflammation, and fibrosis.

### 5.5. Hypertension

Hypertension is characterized by elevated systemic circulation arterial pressure (systolic/diastolic blood pressure), which is accompanied by organ damage. In addition, the physiopathology of hypertension mainly includes hemodynamic changes, endothelial dysfunction, arterial stiffness, and autonomic dysregulation [131].

High-throughput sequencing indicated that spontaneous hypertension in mammals is regulated by m6A-mediated epigenetic transcriptomics [132]. In the genome-wide association studies of GWAS 2011 and GWAS 2018, a total of 1236 m6A-related single nucleotide polymorphisms (M6A-SNPs) were associated with blood pressure (BP) [133]. Another group reported that FTO rs9939609 is positively correlated with melanocortin 4 receptor (MC4R) rs17782313 in hypertensive patients, but MC4R rs17782313 is negatively correlated with diastolic blood pressure and mean blood pressure in male patients [134]. Furthermore, hypertension activates pro-inflammatory macrophages, GLUT1-mediated glycolysis, and PPP flux through mechanisms of tissue hypoxia, endothelial injury, and mitochondrial dysfunction, which may be related to the potential of m6A modification [135]. These results suggested that m6A modification may function as a new mediator of blood pressure regulation, but the detailed underlying mechanism is still far from being elucidated.

### 5.6. Atherosclerosis (AS)

AS is a chronic inflammatory disease characterized by plaques composed of lipid and fiber components that are deposited on the inner wall of blood vessels and, thus, impede blood flow [136]. In addition to clogging blood vessels and causing ischemic damage, plaques rupture and cause heart or brain infarcts. Hence, the development of AS is closely related to inflammatory cell infiltration and immune response, including vascular endothelial cells, macrophages, and VSMCS.

The m6A expression profile of human coronary artery smooth muscle cells (HCASMCs) revealed that METTL3 is upregulated during the proliferation and migration of HCASMCs, and METTL3 is involved in the pathogenesis of AS by affecting protein synthesis and energy metabolism [137]. Previous studies have demonstrated that METTL3-mediated methylation is involved in the inflammatory cascade in endothelial cells (ECs) and is closely associated with hemodynamics and AS formation [138]. In ox-LDL-induced HUVEC and AS mouse models, METTL3/IGF2BP1 promoted HUVEC proliferation, migration, angiogenesis, and AS progression in vivo by upregulating the JAK2/STAT3 pathway [139]. METTL14 promotes endothelial inflammation and atherosclerotic plaque formation by mediating m6A modification of forkhead box O1 (FOXO1) [140], and ischemia-induced ALKBH5 facilitates endothelial angiogenesis by mediating demethylation of sphingosine kinase 1 (SPHK1) [141]. Interestingly, the significantly upregulated METTL14-m6A-miR-19a axis in AS further enhances the proliferative and invasive ability of ASVECs [142]. In CHD and LPS-stimulated THP-1 cells, METTL14 promotes macrophage inflammation and AS development through NF-κB/IL-6 signaling [143]. METTL3 inhibits the expression of circ_0029589 by promoting its m6A modification, which induces macrophage pyroptosis and inflammation in AS [144].

## 6. Conclusions and Perspectives

Recently, conserved, abundant, and reversible m6A methylation has gradually emerged as a critical regulator for gene expression at the post-transcriptional level, which is widely involved in various biological processes, including glycolipid metabolism and metabolic disorders. Extensive studies have established the close association of m6A modification with glucose and lipid metabolism-related enzymes, transcription factors, and signaling pathways. In this review, we comprehensively summarized the pivotal role of m6A modification in the regulation of glycolipid metabolism and the development of GLMD (Table 2), which not only deepens the understanding of the complexity of the regulatory mechanism of maintaining the glucose and lipid homeostasis, but also provides novel insights for the diagnostic and therapeutic strategy of GLMD.

In one-carbon metabolism, S-adenosylmethionine (SAM) serves as a universal methyl donor for m6A modification, and the SAM/S-adenosylhomocysteine (SAH) ratio correlates with the methylation reactions. SAM consumption regulates aerobic glycolysis by activating the EGFR-STAT3 axis and results in increased oxidative stress and a decline in fatty acid catabolism [145,146]. Moreover, SAM is associated with the alterations of hepatic oxidative stress, gluconeogenesis, and fat mass. Methionine depletion mitigates liver steatosis by negatively regulating gluconeogenesis and oxidative phosphorylation, which is associated with NAFLD [147]. Folate consumption leads to hepatocyte degeneration and lipid disturbances by reducing the protein and activity level of Methylenetetrahydrofolate reductase (MTHFR) [148]. Methionine/SAM/folate mediates the transfer of one carbon unit and sources of m6A modification, while glycolipid metabolism and gene regulation are related to methyl donors. Mechanistically, methionine/SAM/folate function upstream of m6A, as well as regulate glycolipid metabolism and GLMD via one-carbon metabolism and methyl transfer.

Nonetheless, alterations of the m6A transcript profile play a dominant role rather than a general role in the complex metabolic network, compared with other genetic regulatory mechanisms such as N1-methyladenine (m1A), 5-methylcytosine (m5C), 7-methylguanosine (m7G), and acetylation modification. Furthermore, previous studies have demonstrated how m6A methylation plays a crucial role in glycolipid metabolism by affecting the mRNA metabolism of target genes and lack of specific regulatory mechanisms, which include: effective screening of target genes, identification of reader proteins, and determination of binding sites by methylated RNA immunoprecipitation (MeRIP), site-directed mutagenesis, and dual-luciferase reporter assay. Meanwhile, the individual variability and tissue specificity of GLMD present a significant challenge to the m6A regulatory network, as few research findings are applied as potential targets, diagnostic basis, or therapeutic means in clinical practice. In summary, the association between m6A modification and glycolipid metabolism provides a basis for exploring the underlying mechanisms of GLMD, but the leading role, detailed mechanism, and clinical application still require further validation.

## Figures and Tables

**Figure 1 biomolecules-13-00273-f001:**
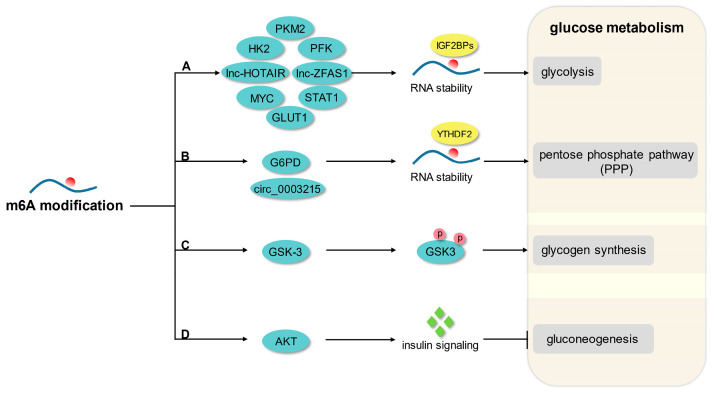
The principal mechanism of m6A modification in glucose metabolism. (A) M6A modification promotes glycolysis by enhancing the mRNA stability of GLUT1, HK2, PFK, STAT1, PKM2, MYC, lnc-HOTAIR, and lnc-ZFAS1. (B) M6A modification enhances PPP flux by mediating the mRNA stability of G6PD and circ_0003215. (C) M6A modification regulates glycogen synthesis by mediating phosphorylation of GSK-3. (D) M6A modification modulates gluconeogenesis in response to AKT-mediated insulin signaling.

**Figure 2 biomolecules-13-00273-f002:**
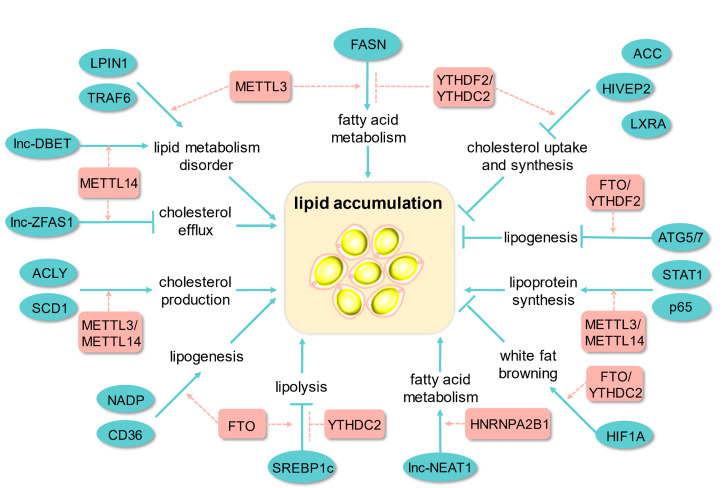
The effects of m6A methylation on lipid accumulation. FASN, fatty acid synthase; ACC, Acetyl-CoA carboxylase; HIVEP2, HIVEP zinc finger 2; LXRA, liver X receptor; ATG5/7, autophagy related 5/7; HIF1A, hypoxia inducible factor 1 subunit alpha; SREBP1c, sterol regulatory element binding protein-1 c; NADP, nicotinamide adenine dinucleotide phosphate; CD36, CD36 molecule; ACLY, ATP citrate lyase; SCD1, stearoyl-CoA desaturase 1; STAT1, signal transducer and activator of transcription 1; LPIN1, lipin 1; TRAF6, TNF receptor-associated factor 6.

**Figure 3 biomolecules-13-00273-f003:**
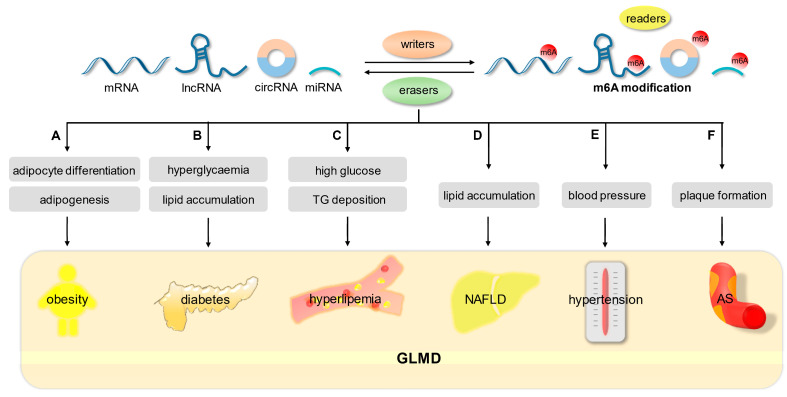
The role of m6A in GLMD by regulating glycolipid metabolism. (A) The m6A modification is closely associated with obesity through the regulation of adipocyte differentiation and adipogenesis. (B) M6A modification regulates diabetes progression by affecting glucose homeostasis and lipid accumulation. (C) M6A methylation modulates hyperlipemia by affecting glucose and TG metabolism. (D) M6A modification is involved in hepatic lipid accumulation in NAFLD. (E) The relationship between blood pressure and m6A-SNP. (F) M6A modification regulates plaque formation in AS.

**Table 1 biomolecules-13-00273-t001:** The biological function of m6A regulators in RNA metabolism.

Type	Regulator	Biological Function	References
m6A writers	METTL3	Catalyzes m6A modification	[16]
	METTL14	Assists METTL3 to recognize the specific subtract and enhances the stability of MTC structure	[16]
	METTL16	Catalyzes m6A modification and regulates SAM homeostasis	[17]
	WTAP	Promotes the cellular m6A deposition	[18]
	VIRMA (KIAA1429)	Guides the core component of MTC to specific RNA region and interacts with cleavage and polyadenylation specific factor 5/6 (CPSF5/6)	[18]
	RBM15/15B	Recruits METTL3-METTL14 heterodimer to specific RNA sites	[19]
	ZC3H13	Contributes to the nuclear localization of MTC	[20]
	IME4	Mediates m6A modification of yeast mRNA	[21]
	MUM2	Mediates m6A modification of yeast mRNA	[21]
m6A erasers	FTO	Removes m6A modification to promote mRNA splicing and translation	[22]
	ALKBH5	Removes m6A modification to promote mRNA splicing, mRNA nuclear output, and long 3′-UTR mRNA production	[23]
m6A readers	YTHDF1	Promotes mRNA translation and protein synthesis	[24]
	YTHDF2	Reduces mRNA stability and regulates mRNA localization	[24]
	YTHDF3	Interacts with YTHDF1 to promote mRNA translation or assists YTHDF2-mediated RNA degradation	[24]
	YTHDC1	Contributes to RNA splicing, export, and transcriptional silencing	[25]
	YTHDC2	Promotes the translation of target RNA but reduces their abundance	[26]
	eIF3	Promotes mRNA translation	[27]
	IGF2BPs	Enhances the stability and translation of target RNA	[28]
	HNRNPA2B1	Regulates alternative splicing and primary microRNA processing	[29]
	HNRNPC/G	Mediates mRNA splicing	[29]

**Table 2 biomolecules-13-00273-t002:** Multiple functions of m6A regulators in GLMD.

Disease	Regulator	Target	Function	References
Obesity	FTO	SRSF2, RUNX1T1	FTO depletion induces preadipocyte differentiation by regulating the RNA binding ability of SRSF2 and exonic splicing of RUNX1T1	[99]
	FTO	PPARG	FTO promotes the differentiation of BMSCs into adipocytes by increasing the expression of PPARG	[100]
	YTHDF1	MTCH2	YTHDF1 facilitates MTCH2 mRNA translation and adipogenesis	[102]
	METTL3/YTHDF2	CCND1	METTL3/YTHDF2 mediated CCND1 degradation inhibits cell-cycle progression and adipogenesis	[103]
DM	FTO	FOXO1, G6PC, DGAT2	FTO promotes the expression of FOXO1, G6PC and DGAT2, which results in hyperglycemic phenotype	[106]
	METTL3	FASN	METTL3 facilitates fatty acid metabolism and insulin resistance by increasing the expression of FASN	[75]
	YTHDC2	FASN, ACC, SREBP-1c, SCD-1	YTHDC2 suppresses liver steatosis by decreasing of mRNA stability of lipogenic genes	[78]
	IMP2/IGF2BP2	PDX1	IMP2/IGF2BP2 promotes pancreatic β-cell proliferation and insulin secretion by enhancing PDX1 expression	[107]
	METTL3/ METTL14	Pdx1, MafA, Nkx6.1, GLUT2	METTL3/14 promotes β-cell proliferation and functional maturation by increasing the expression of Pdx1, MafA, Nkx6.1, and GLUT2	[108,109]
	METTL14/ YTHDF2	LncRNA TINCR	METTL14/YTHDF2 suppresses pyroptosis and diabetic cardiomyopathy by mediating the degradation of lncRNA TINCR	[111]
	IMP2	p53	LncRNA Airn prevents the development of cardiac fibrosis in the diabetic heart via the IMP2-p53 axis	[112]
	METTL3	LncRNA SNHG7	METTL3 regulates endothelial-mesenchymal transition in diabetic retinopathy by enhancing the stability of lncRNA SNHG7	[113]
Hyperlipemia	METTL3	PRKD2	METTL3 inhibits PRKD2 expression and the activity of GLUT4/IRS-1/AKT signaling, which results in high levels of glucose and TG	[115]
	FTO	CREB, AMPK	FTO modulates the activity of the CREB signaling pathway and AMPK pathway	[119,120]
NAFLD	METTL3/YTHDF1	Rubicon	METTL3/YTHDF1 promotes Rubicon expression and hepatic lipid deposition	[123]
	FTO	SREBF1, SCD1	FTO induces lipid accumulation by increasing the expression of SREBF1 and SCD1	[124]
	FTO	IL-17	FTO promotes IL-17 expression and chronic liver inflammation	[125]
	ALKBH5	LINC01468	ALKBH5 drives lipogenesis and NAFLD-HCC progression by increasing the RNA stability of LINC01468	[126]
	YTHDF3	PRDX3	YTHDF3-mediated PRDX3 translation alleviates liver fibrosis	[127]
	ALKBH5	PTCH1	ALKBH5 ameliorated liver fibrosis and suppressed HSCs activation via triggering PTCH1 activation	[128]
Hypertension	FTO rs9939609	MC4R rs17782313	FTO rs9939609 is positively correlated with MC4R rs17782313, which is inversely correlated with diastolic blood pressure in male patients	[134]
AS	METTL14	lncRNA ZFAS1	METTL14 limits cholesterol efflux by mediating the m6A modification of lncRNA ZFAS1	[87]
	METTL3	PI3K/AKT pathway	METTL3 facilitates HCASMC proliferation and migration by activating PI3K/AKT pathway	[137]
	METTL3/IGF2BP1	JAK2/STAT3 pathway	METTL3/IGF2BP1 promoted HUVEC proliferation, migration, and angiogenesis by positively regulating JAK2/STAT3 pathway	[139]
	METTL14	FOXO1	METTL14 aggravates endothelial inflammation and AS by increasing the expression of FOXO1	[140]
	ALKBH5	SPHK1	ALKBH5 helps in the maintenance of angiogenesis in endothelial cells following acute ischemic stress via increased SPHK1 expression	[141]
	METTL14	miR-19a	METTL14 enhances the proliferation and invasion of ASVEC by promoting the processing of mature miR-19a	[142]
	METTL14	NF-κB/IL-6 pathway	Mettl14 mediates the inflammatory response of macrophages through the NF-κB/IL-6 pathway	[143]
	METTL3	circ_0029589	METTL3 induces macrophage pyroptosis and inflammation by inhibiting circ_0029589	[144]

## Data Availability

Not applicable.

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
