# Peer review of "The Epigenetic Regulation of RNA N6-Methyladenosine Methylation in Glycolipid Metabolism"

_biomolecules, 2023, doi:10.3390/biom13020273_

Round 1

Reviewer 1 Report

This review by the authors of metabolic drivers of m6A modifications and their links to metabolic disease is a largely comprehensive review of the literature. 

There is no discussion of m6A modification and SAM/folate metabolism/methionine deprivation. As all of these have links to metabolic outcomes and m6A frequency, it seems like this should be addressed. 

It would also be nice to see described in the sections on disease how the disease alters the metabolic pathways described in the above sections and the potential to impact m6A modification beyond just alterations of genes associated with disease states.

Reviewer 2 Report

In this review, Dr. Yang and colleagues discussed the regulation and roles of RNA modifications in regulating glycolipid metabolism and the development of glycolipid metallic disease (GLMD). In general, this manuscript covers most of the studies in this area and is of great interest and significance as RNA modification studies are emerging. Several comments need to be addressed.

1. The current review is only focus on coding mRNAs. As the author mentioned, m6A modification can exsit and plays esseatial role in both coding mRNA and noncoding RNAs (miRNAs and long noncoding RNAs), the description of nRNA modification and if it regulates glycolipid metabolism and/or is involved in GLMD should also be included.

2. In table 1, only general roles of m6A modification-related proteins are listed. It will be more informative if the specific roles of m6A regulators in glycolipid metabolism, if any, could be discussed.

3. The molecular mechanisms by which m6A controls glycolipid metabolism is not well discussed and displayed. It will be more clear if a summary table could be included showing the essential genes regulated by m6A, the effect of m6A on those gene’s activities and if the dyregulation of genes is linked to any GLMD, et al. Also, in addition to controlling gene expression, the other effect of m6A on gene activity should be included.

Round 2

Reviewer 2 Report

In the revised version, the authors made some progression. I have some minor comments:

1. A lot of references are missing, such as lines 54-59. The authors should carefully revisit the whole manuscript and add proper references.

2. The function of m6A-noncoding RNA glycolipid metabolism should also be reflected in the figs. For example, fig3 only indicates the m6A of mRNA, however, actually, lncNEAT1 also regulates adipogenesis as demonstrated by authors. This bias of only mRNA in figs and tables should be avoided.

Author Response

Dear Reviewer,

Thank you very much for your constructive comments and recommendations. We have addressed them accordingly in the revised version of the manuscript entitled: “The Epigenetic regulation of RNA N6-methyladenosine methylation in glycolipid metabolism” as described in details below. We hope that our revisions improved the paper such that Reviewer and Editor now deem it suitable for publication in Biomolecules.

Below please find the detailed responses to Reviewer’s comments:

Point 1: A lot of references are missing, such as lines 54-59. The authors should carefully revisit the whole manuscript and add proper references.

Our reply: Thank the reviewer very much for the advice. As the reviewer pointed out, we have revised the whole manuscript and added proper references (lines 56-57). To prevent the loss of references, we have submitted the revised version in PDF format (please see the attachment).

New addition in the manuscript that reads:

lines 56-57: “Other than mRNA, m6A occurs in long non-coding RNA (lncRNA), circular RNA (circRNA) and microRNA (miRNA) [13-15].”

Point 2: The function of m6A-noncoding RNA glycolipid metabolism should also be reflected in the figs. For example, fig3 only indicates the m6A of mRNA, however, actually, lncNEAT1 also regulates adipogenesis as demonstrated by authors. This bias of only mRNA in figs and tables should be avoided.

Our reply: Thank the reviewer very much for the suggestion. We have modified figure 1, 2 and 3 to visualize the the relationship of nRNA modification in glycolipid metabolism and GLMD. Moreover, we have supplemented table 2, which summarizes the m6A modification of mRNA, lncRNA, circRNA and miRNA in GLMD.
